# Biological Invasions Affect Resource Processing in Aquatic Ecosystems: The Invasive Amphipod *Dikerogammarus villosus* Impacts Detritus Processing through High Abundance Rather than Differential Response to Temperature

**DOI:** 10.3390/biology12060830

**Published:** 2023-06-07

**Authors:** Benjamin Pile, Daniel Warren, Christopher Hassall, Lee E. Brown, Alison M. Dunn

**Affiliations:** 1School of Biology, University of Leeds, Leeds LS2 9JT, West Yorkshire, UK; c.hassall@leeds.ac.uk (C.H.); a.dunn@leeds.ac.uk (A.M.D.); 2Animal and Plant Health Agency (APHA), Sand Hutton YO41 1LZ, York, UK; daniel.warren@apha.gov.uk; 3School of Geography and Water@Leeds, University of Leeds, Leeds LS2 9JT, West Yorkshire, UK; l.brown@leeds.ac.uk

**Keywords:** invasive alien species, temperature, parasite, shredding, density, impact

## Abstract

**Simple Summary:**

Ecosystems are affected by multiple stressors, which interact in ways that can be difficult to predict. Stressors such as climate warming and introductions of non-native species and parasites can impact processes vital to the functioning of ecosystems. In temperate freshwater ecosystems most nutrients originate from leaf litter, which is processed by invertebrate shredder species. Focusing on the invasive killer shrimp, which is replacing native species, we investigated how the stressors interact to impact rates of shredding and survival of the shredder species to make predictions of how temperature and invasive species may alter the function of temperate freshwater ecosystems. Increasing temperature was found to increase rates of shredding up to an optimum, after which shredding decreased. The native shredders had a higher rate of shredding than the invasive species at all temperatures. However, the invasive killer shrimp reached a much higher abundance than the native and we demonstrated that the total population of the introduced shredders processed far more leaf litter at invaded sites. While this may make the ecosystems more productive in the short term, it may lead to the exhaustion of the leaf litter resource, with negative consequences for the function of the ecosystem over time.

**Abstract:**

Anthropogenic stressors such as climate warming and invasive species and natural stressors such as parasites exert pressures that can interact to impact the function of ecosystems. This study investigated how these stressors interact to impact the vital ecosystem process of shredding by keystone species in temperate freshwater ecosystems. We compared metabolic rates and rates of shredding at a range of temperatures up to extreme levels, from 5 °C to 30 °C, between invasive and native amphipods that were unparasitised or parasitised by a common acanthocephalan, *Echinorhynchus truttae*. Shredding results were compared using the relative impact potential (RIP) metric to investigate how they impacted the scale with a numerical response. Although per capita shredding was higher for the native amphipod at all temperatures, the higher abundance of the invader led to higher relative impact scores; hence, the replacement of the native by the invasive amphipod is predicted to drive an increase in shredding. This could be interpreted as a positive effect on the ecosystem function, leading to a faster accumulation of amphipod biomass and a greater rate of fine particulate organic matter (FPOM) provisioning for the ecosystem. However, the high density of invaders compared with natives may lead to the exhaustion of the resource in sites with relatively low leaf detritus levels.

## 1. Introduction

Ecosystems worldwide are subject to impacts from multiple abiotic and biotic stressors, which stem from a combination of natural and anthropogenic drivers [1]. Abiotic factors such as climate warming, ocean acidification, pollution and land-use change act by influencing species’ physiology and extirpate those organisms for which conditions shift beyond physiological limits. Biotic stressors such as species introductions, predation, parasitism, disease and spatiotemporal decoupling from food resources act at a population level to influence demographic processes [2,3,4]. To reflect the complexity of natural systems, more research is needed on the effects of multiple factors in synergy, rather than considering the impacts of single stressors in isolation [5,6,7].

Freshwater habitats are among the most threatened by interacting stressors. These ecosystems are biodiverse, occupying only 0.8% of the world’s surface and 0.01% of the world’s water, but harbouring 10% of described species [8,9]. This biodiversity supports high productivity and provides important resources to adjacent ecosystems and humankind [10,11]. In temperate freshwater ecosystems, the basal energy resource is often heavily skewed towards allochthonous riparian leaf litter [12] with macroinvertebrate shredders contributing to the release of nutrients, dispersing shredded leaf particulates and transferring biomass up through trophic webs [13,14,15]. However, research is required into how this shredding may be impacted by multiple stressors [7,16,17].

Among the most significant biotic stressors facing natural systems are invasive species [18,19,20]. Successful invaders often outcompete native species by monopolising resources and they may also prey upon natives, rapidly altering the community structure [21,22,23]. Alterations of this type may affect ecosystem functions and resource processing, causing cascading effects across trophic levels [24,25,26]. Amphipods are important shredder species in temperate freshwaters [27,28], but those from the Ponto-Caspian region such as the ‘killer shrimp’ *Dikerogammarus villosus* (Sowinsky, 1894) have proved to be highly invasive. *D. villosus* has invaded major waterways throughout western Europe, has been present in the UK since at least 2010 and is predicted to invade North America and Ireland [23,29,30]. This invasion has resulted in the replacement of native amphipods and impacted wider macroinvertebrate communities through competition and predation [29,31,32]. *Gammarus pulex* (Linnaeus, 1758) is the dominant native freshwater amphipod shredder in Great Britain, but it is outcompeted and preyed on by the invasive *D. villosus*, leading to species replacement [32,33,34,35,36].

In addition to anthropogenic stressors, species face a suite of stressors as a result of their place within ecological networks of interactions. Parasites are a ubiquitous part of natural communities that may alter patterns of host survival and can also affect host traits, including behaviour and feeding rates [37,38,39,40]. The native amphipod *G. pulex* is commonly parasitised by acanthocephalans, including *Echinorhynchus truttae* (Schrank, 1788), which uses the amphipod as an intermediate host and can have a prevalence of up to 70% in host populations [41]. Parasite manipulation by *E. truttae* alters both anti-predator avoidance [41,42,43,44] and the feeding behaviour of the amphipod host [40,45].

Alterations to host feeding behaviours may be due to energetic costs exerted by parasites and reflected in changes to metabolic rates. The effect of the acanthocephalan infection of amphipods by *Polymorphus minutus* and *Pomphorhynchus laevis* on the basal metabolic rate (BMR) has previously been demonstrated, but *E. truttae* has not previously been studied in relation to its effect on the host BMR [46,47]. In contrast to the native species, *D. villosus* has benefitted from enemy release in much of its new range, with some parasites from its native range absent from invader populations and no evidence of infection by *E. truttae* in invaded areas [48,49].

Climate change has long been established as a threat to biodiversity due to higher mean temperatures as well as an increase in the frequency and intensity of extreme events [50,51]. High temperatures experienced during heatwaves can impact on the structure and function of communities [52,53]. Climate change may also make ecosystems more vulnerable to the impact of additional stressors such as invasive species [54,55,56], which may have different thermal optima than their native analogues. Tolerance to a wide range of conditions is a characteristic of successful invasive species, facilitating establishment and spread; however, survival and behaviour become less predictable at high temperatures [57,58,59]. Experimental work is, therefore, required to investigate how invasive species survive and function in high temperatures in comparison with native analogues. Previous studies comparing shredding rates of *G. pulex* and *D. villosus* reported a positive correlation between shredding rates and temperature, but comparisons between the native and invader conflicted [60,61]. Interactions between parasites and hosts can also be influenced by temperature, with outcomes difficult to predict when the thermal optima of hosts or parasites are exceeded due to species-specific interactions [62,63,64,65]. Previous studies have not considered the impact of extreme temperatures and of parasitic infections on shredding efficiency.

This study investigated how survival and shredding differed between native and invasive amphipods as a function of two significant stressors: temperature and a common parasite of native amphipods, *E. truttae*. We hypothesised that the invasive *D. villosus* would outcompete the native *G. pulex* partly through higher rates of shredding, which may be underpinned by differences in metabolic rates. The metabolic theory of ecology (MTE) proposed that the rates of almost all activities undertaken by biological organisms—for example, survival, growth, development and reproduction—are determined by metabolic rates. Metabolic rates, the rates of energy uptake, processing and allocation, are determined by other factors in turn. One of the main factors affecting an individual’s metabolic rate is temperature [66,67]. As temperature rises, the rates of metabolic reactions within organisms increase, raising demands for resource acquisition. Previous studies have shown both *D. villosus* and *G. pulex* increased their shredding efficiency at higher temperatures [60], but they did not directly measure metabolic rates. Additionally, interspecific biological interactions have been shown to influence temperature effects on metabolic scaling in some amphipods [68,69], but the effects of parasite infections on the relative performance of *G. pulex/D. villosus* have not yet been considered.

The native species *G. pulex* suffers more parasitism than the invasive *D. villosus* in Great Britain [70], so the impact of the common parasite *E. truttae* was tested to determine whether the metabolic demands of the parasite altered the behaviour and survival of the native amphipod. The lesser competitive power of the native was hypothesised to be exacerbated by parasitism. We aimed to identify whether species and parasitise status interacted with temperature to alter survival and behaviour to investigate whether multiple stressors could have amplified impacts on the ecosystem process of shredding. We also used the relative impact potential (RIP) metric to compare shredding data in respect of a numerical response due to the high densities in which the invasive *D. villosus* is found at invaded UK sites in order to test the hypothesis that the high densities of the invasive species would lead to a higher potential impact for *D. villosus* [71,72].

## 2. Materials and Methods

*Gammarus pulex* were collected by kick-sampling from Meanwood Beck at Golden Acre Park (53.8687° N, −1.5884° E) and Meanwood Park, West Yorkshire (53.8301° N, −1.5746° E), UK. *Dikerogammarus villosus* were collected from Grafham Water, Cambridgeshire (52.2909° N, −0.0323° E), UK. The mean freshwater temperature in the UK is 11 °C, with a 95th percentile minimum and maximum of 2.5 °C and 20.3 °C, respectively [73], although the shallowness of the Meanwood Beck site may have led to significantly lower minimum temperatures and higher maximum temperatures; however, recorded temperature data were unavailable. All animal collections were carried out between February and April, when mean UK river temperatures are approximately 8.20 °C [74]. All amphipods were kept for a minimum of 5 days prior to experiments in order to acclimate to laboratory conditions in species-specific communal tanks in a controlled temperature room maintained at 15 ± 0.1 °C SD with a 12:12 h light/dark cycle (08:00–20:00) (Figure 1). The animals were not sexed, but were roughly size-matched, ensuring that all were at the same mature life stage.

Tanks were filled with aerated aged tap water and the study organisms were fed ad libitum with leaf litter. Parasitised animals were identified by a visual examination. Individuals parasitised by *E. truttae* were identifiable by an orange acanthocephalan cystacanth visible through the cuticle [75]. Only hosts with a mature cystacanth were used, which facilitated pre-experimental identification. The infection status was confirmed after experiments by dissection and a visual identification [76].

The leaf material selected for experimental use was common alder (*Alnus glutinosa*) as amphipod species have demonstrated a preference for the leaf detritus of this species [77]. Alder is a nitrogen-fixing species, producing nutrient-rich leaf litter relatively low in carbon, which is favoured by detritivorous species in fresh waters [77,78]. Alder is also a common riparian species throughout temperate areas of the northern hemisphere [79] and is found at the locations where the experimental animals were collected. Leaves were collected as natural autumn leaf fall and dried. Leaves were then conditioned for two weeks in water from Meanwood Beck, West Yorkshire, to promote microbial colonisation and to increase the palatability of the detritus for amphipods [77]. Once conditioned, a cork-borer was used to cut 6 mm diameter discs of leaf, avoiding the lignified and less palatable midrib and veins. Leaf discs were air-dried and then weighed out in sets of 15 (mean mass = 27.0 ± 1.9 mg SD) and subsequently conditioned in Meanwood Beck stream water for 48 h immediately prior to the experimental period. The drying of the discs prior to weighing killed the microbes that had accumulated as a biofilm during the initial leaf conditioning while leaving a nitrogen-rich mass. The reconditioning was carried out so that some live microbial film was present when fed to the experimental animals in addition to the previously accumulated nitrogen-rich biomass.

The leaf shredding and survival rates of *D. villosus* and *G. pulex* that were either unparasitised or parasitised by *E. truttae* were measured at a range of temperatures. Survival and shredding data were derived from the same experiment, with mortality being recorded if suffered during the shredding experiment. Amphipods were roughly size-matched in order to ensure all were at the same mature life stage to minimise possible differences in the mass-specific thermal sensitivity of metabolic rates [80]. Animal sizes were representative of mature individuals found at the field sites, with some selection for a size match. Individual amphipods were placed on a paper towel to remove excess water and weighed (unparasitised *G. pulex* mean mass was 0.050 ± 0.012 g, parasitised *G. pulex* mean mass was 0.031 ± 0.011 g and *D. villosus* mean mass was 0.063 ± 0.015 g). Animals were individually placed in transparent, circular plastic containers (diameter of 7 cm and depth of 5 cm) with 250 mL of aged tap water. Two transparent glass beads were placed in the containers to provide a refuge and to prevent excess swimming due to thigmotactic behaviours [81,82] while still allowing observations (Figure 2). The containers were then placed in incubators at 15 °C with a 12:12 h light/dark cycle and the animals underwent a 24 h starvation period to standardise hunger, during which the temperature was gradually increased or decreased at a rate of 1 °C every 2 h until the desired temperature for the treatment was reached. Temperature treatments were between 5 and 30 °C in 5 °C increments with the following number of replicates for the initial survival experimental data: 5 °C (unparasitised *G. pulex n* = 16, parasitised *G. pulex n* = 13 and *D. villosus n* = 16), 10 °C (unparasitised *G. pulex n* = 16, parasitised *G. pulex n* = 14 and *D. villosus n* = 16), 15 °C (unparasitised *G. pulex n* = 15, parasitised *G. pulex n* = 16 and *D. villosus n* = 16), 20 °C (unparasitised *G. pulex n* = 16, parasitised *G. pulex n* = 15 and *D. villosus n* = 16), 25 °C (unparasitised *G. pulex n* = 16, parasitised *G. pulex n* =14 and *D. villosus n* = 16) and 30 °C (unparasitised *G. pulex n* = 16, parasitised *G. pulex n* = 16 and *D. villosus n* = 16).

After the starvation period, 15 weighed leaf discs were added to each container. Animals were checked every 24 h and mortality was recorded. Mortality was identified as a lack of pleopod beating and the absence of a reaction to a physical stimulus. These treatments were selected to test outcomes at a range of temperatures up to and beyond known thermal limits. Water levels in each container were maintained, with oxygenated water at the relevant temperature being added if required.

After 6 days, the experiment was halted and any remaining leaf discs, hereon identified as coarse particulate organic matter (CPOM), were stored in ethanol to stop further microbial decomposition before reweighing. For the measurement of CPOM samples, metal weighing boats were heated in a drying oven for 24 h at 60 °C and weighed. Individual CPOM samples were allocated to a weighing boat to then be heated in the drying oven for 24 h at 60 °C. Boats and leaves were then weighed together and the mass of CPOM was calculated.

The metabolic rates of *D. villosus* (unparasitised) and *G. pulex* (unparasitised/parasitised) were measured at 10 °C (unparasitised *G. pulex n* = 17, parasitised *G. pulex n* = 17 and *D. villosus n* = 12), 20 °C (unparasitised *G. pulex n* = 21, parasitised *G. pulex n* = 11 and *D. villosus n* = 18) and 25 °C (unparasitised *G. pulex n* = 17, parasitised *G. pulex n* = 16 and *D. villosus n* = 20). Separate subsets of animals were used for the metabolic rate and survival/shredding experiments. Animals were size-matched as much as possible to minimise differences in the mass-specific thermal sensitivity of metabolic rates [80]. Animal sizes were representative of mature individuals found at the field collection sites but with some selection for a size match (unparasitised *G. pulex* mean mass was 0.041 ± 0.010 g, parasitised *G. pulex* mean mass was 0.033 ± 0.008 g and *D. villosus* mean mass was 0.072 ± 0.020 g). Animals were individually placed in plastic containers as used in the shredding experiment above with glass beads and aged tap water, with the temperature gradually changed as required. The animals underwent a 24 h starvation period, ensuring that measurements were post-absorptive and unaffected by metabolism of food [83]. Amphipods were dabbed dry on paper towels before being weighed and placed in a closed-circuit respirometry vial (diameter of 15 mm, height of 48 mm and volume of 4 mL; © OXVIAL4) containing fully aerated water at the relevant temperature. A small section of plastic mesh was also inserted into the vial to restrict amphipod movement and encourage natural clinging and resting behaviours to allow the measurement of the basal metabolic rate [84]. The amphipods were acclimated for 30 min before measurements were taken. An optical oxygen sensor (Pyroscience© Piccolo2) was used to measure the dissolved oxygen content of the water in mg/L (ppm) at the beginning of the test and again after a 30 min period, giving a decrease in milligrams of oxygen per litre of water. This figure was converted to a rate by calculating the reduction in oxygen per hour adjusted for amphipod size to give a rate of oxygen consumption per gram of amphipod.

The impact of the invasive species and parasitism on resource processing depends on both the leaf shredding capability and relative abundance. The RIP metric incorporates the relative consumer abundance response as a means of scaling relative per capita effects to compare the relative impact potential of invasive versus native species. We applied the RIP to test the hypothesis that the high densities of the invasive species would lead to a higher impact potential for *D. villosus*. The metric was used to compare the relative impact potential of these freshwater amphipod species using abundance data and the results of the survival/shredding experiment. To calculate the relative impact potential, the functional response asymptotes or maximum feeding rates of the native and invasive species as well as their relative abundances were compared.
$$\mathrm{RIP}=\left(\frac{\mathrm{Invader}\;\mathrm{FR}}{\mathrm{Native}\;\mathrm{FR}}\right)\times \left(\frac{\mathrm{Invader}\;\mathrm{abundance}}{\mathrm{Native}\;\mathrm{abundance}}\right)$$

When the result of this equation is >1, the invader can be regarded as having a greater potential impact on the invaded ecosystem than that exerted by the native resident. The higher the RIP score, the higher the impact of the invasive species relative to the native. The RIP metric has previously been extensively used to compare the relative impact of invasive predators [72] and of algal uptake by filter feeders [85]. Here, we applied this metric for the first time to explore the impact of biological invasion on the key process of leaf shredding. As leaf detritus was supplied in excess, the feeding rate on leaf matter was used as the measure of consumption representative of the functional response curve asymptote maximum feeding rate. Abundance data were taken from Warren et al. [76], a study that was undertaken using the same field sites as this study, with additional data on parasite prevalence calculated as a percentage of the animals collected for this study.

### Data Analysis

All analyses were produced using R, with plots for shredding, metabolic rates and relative impact potential created using the “ggplot2” package [86,87]. Due to the significantly higher mass of *D. villosus* compared with *G. pulex* (Student’s *t*-test; *t* = −14.36 and *p* < 0.001) and unparasitised *G. pulex* being larger than parasitised conspecifics (*t* = 4.19 and *p* < 0.001), data were standardised by body mass (g). A general additive model (GAM) was constructed using the “mgcv” package to assess the effect of amphipod species, temperature and parasitised status on the rates of shredding [88]. Temperature was modelled using a tensor smooth, which improved the model fit, and the mass of leaf consumed per gram of amphipod was transformed using the natural log, which reduced heteroscedacity and improved the residual distribution.

Survival statistics were modelled using a Cox proportional hazards model, with plots produced using Kaplan Maier product limit estimator curves using the “survival” package in R [89]. To determine the activation energy of shredding efficiency, temperature was standardised to 1/*k*T_c_–1/*k*T, with *k* representing the Boltzmann constant (8.62 × 10^−5^ eV K^−1^), T representing temperature in ° Kelvin and c representing the allocated intercept temperature of 15 °C or 288.15 °K. A linear regression model was used to analyse the ln-transformed rates of grams of leaf shredded per gram of amphipod per day against a standardised temperature [61,90]. Data for shredding rates at 5, 10 and 15 °C were used for this analysis to enable comparisons with Kenna et al. [61], who used the same experimental set-up but a different resource (*Acer pseudoplatanus* leaves). ANCOVA was used to compare shredding rates between amphipod treatments with pairwise Tukey’s post hoc tests.

The metabolic rate was calculated by measuring milligrams of oxygen consumed per hour per gram of subject amphipods. To test for differences in metabolic rates between amphipod species and parasitised status, a Quade’s ANCOVA was carried out with Wilcoxon pairwise post hoc tests. A one-way ANOVA was carried out to test for differences in metabolic rates for amphipods between temperature regimes and post hoc Tukey’s tests were used for pairwise comparisons of metabolic rates between temperatures.

In the field, the abundance of *G. pulex* parasitised by *E. truttae* was extremely low, at a mean of 4.38 individuals per m^2^ compared with 164 per m^2^ for unparasitised individuals and 1176 individuals per m^2^ for *D. villosus*. Therefore, due to the low abundance, an RIP analysis was not carried out using parasitised *G. pulex.* The abundance data showed that there were differences in the abundances of the native and the invasive amphipods in the field. Estimates were made based on multiple counts at multiple locations, with mean abundances of 83.280 (±15.710) individuals per m^2^ for *D. villosus* and 17.378 (±4.486) per m^2^ for *G. pulex* [72]. Mortality did not differ between species by temperature treatment; therefore, the abundance data did not need to be adjusted as the ratio of species’ abundances remained the same as both suffered mortality at the same rate under each temperature regime. Variation and uncertainty were accounted for by using standard deviations of all data in probability density functions [71].

## 3. Results

### 3.1. Shredding

The shredding rate significantly differed between amphipod species (F_(1244)_ = 142.30; *p* < 0.001), with *G. pulex* having a higher rate of shredding than *D. villosus* (Figure 3) (Table 1). Temperature had a significant effect on rates of shredding (F_(5240)_ = 76.07; *p* < 0.001). There was no significant interaction between temperature and species (F_(5244)_ = 2.37; *p* = 0.13), but temperature and parasitism status were found to significantly interact to affect rates of shredding (F_(10, 228)_ = 4.25; *p* = 0.02). The shredding rates of unparasitised *G. pulex* and *D. villosus* peaked at 15 and 20 °C, respectively, while the shredding of *G. pulex* infected with *E. truttae* had an accelerating rate of increase as the temperature increased, with a greater variation than the other treatments (Figure 3). Comparing *G. pulex* treatments, there was no difference between parasitised and unparasitised amphipods (F_(1155)_ = 0.65; *p* = 0.42), but parasitised status significantly interacted with temperature (F_(11, 155)_) = 7.50; *p* = 0.01) (Appendix A).

The activation energy of shredding between 5 and 15 °C for unparasitised *G. pulex* was 0.39 eV (95% CI: 0.17–0.60) and *G. pulex* parasitised by *E. truttae* was 0.36 eV (95% CI: 0.09–0.63) (Figure 4). However, the activation energy for *D. villosus* was higher than predicted by the MTE at 0.90 eV (95% CI: 0.72–1.09) (Table 2) [90]. The ANCOVA showed a significant difference between amphipod treatments (F_(2, 134)_ = 9.45; *p* < 0.001), with pairwise Tukey’s post hoc tests finding significant differences between unparasitised *G. pulex* and *D. villosus* (t_(87)_ = 2.40; *p* = 0.05) and between *G. pulex* parasitised with *E. truttae* and *D. villosus* (t_(89)_ = 4.34; *p* < 0.001), but no difference was detected between unparasitised and parasitised *G. pulex* (t_(87)_ = 1.90; *p* = 0.14).

### 3.2. Survival

Temperature had a significant effect on survival (z = 7.79; *p* < 0.001), with a strong negative correlation identified (Figure 5a–c). An increase in temperature was associated with a 1.25 greater hazard of mortality (95% CI: 1.18, 1.32). Survival did not differ between species of amphipods (z = 0.73; *p* = 0.47) and no interaction was found between temperature and species (z = −1.32; *p* = 0.19). Survival over the experimental period did not differ between parasitised and unparasitised *G. pulex* (z = −1.28; *p* = 0.20), but an interaction was detected between temperature and time to mortality for unparasitised *G. pulex* (z = 2.66; *p* = 0.01). There was an increased likelihood of mortality occurring earlier in the *G. pulex* that were not parasitised by *E. truttae* than parasitised conspecifics, with a 1.06 greater hazard of earlier mortality (95% CI: 1.02, 1.11) (Figure 5b).

### 3.3. Metabolic Rate

There was a significant difference in the rate of oxygen consumption between temperature treatments (ANOVA F_(2, 147)_ = 18.36; *p* < 0.001) (Table 3). Post hoc Tukey’s tests found significant differences between all temperature treatments (10 °C, 20 °C and 25 °C) in pairwise tests (10 and 20 °C (*p* < 0.001), 10 and 25 °C (*p* < 0.001) and 20 and 25 °C (*p* < 0.01)). No significant interaction between species and temperature on metabolic rates was found using Quade’s ANCOVA (F_(2, 4)_ = 5.64; *p* = 0.07). Post hoc Wilcoxon pairwise tests revealed no significant differences in metabolic rates between species (*p* = 0.75) or parasitised status (*p* = 1) (Figure 6).

### 3.4. Invader Relative Impact Potential

In all temperature regimes, the invasive *D. villosus* had mean RIP scores > 1 (Table 4). This indicated greater density-scaled shredding rates in the invader relative to the native species, with an increasing RIP score indicating a greater relative shredding rate. There was a general trend of increasing mean RIP scores with increasing temperature, from 4.82 at 5 °C to the highest RIP of 20.85 at 25 °C, with the RIP then falling to 7.56 at 30 °C.

The higher potential impact of *D. villosus* was due to the extremely high densities at which this species was found at the invaded sites compared with the much lower densities at which *G. pulex* was found. Despite the lower maximum shredding rate of the invasive species, the higher abundance of the invader led to higher RIP values, which predicted higher rates of shredding in ecosystems where the invader was present (Figure 7).

## 4. Discussion

The invasive amphipod *D. villosus* is likely to affect shredding rates and, hence, nutrient availability in freshwater systems as a result of its propensity to reach a high abundance and maintain a high density and dominance in invaded communities post-invasion [91]. Although the native species *G. pulex* has a higher per capita shredding rate, it occurs at lower densities than the invasive species in the UK, with the invasive *D. villosus* currently restricted in distribution to a few specific locations [72,92]. The high predatory and competitive abilities [32,36] as well as the enemy release experienced by *D. villosus* in Great Britain [70] facilitates high-density populations, which are likely to alter shredding and the nutrient flow in invaded freshwater systems. Parasitism affected amphipod shredding, with the highest rate of shredding in *G. pulex* parasitised with *E. truttae*. At lower temperatures, parasitised *G. pulex* maintained a similar shredding rate to the other amphipod treatments; however, at 20 °C and above, the shredding rate for parasitised *G. pulex* individuals rapidly increased. This correlated with a previous study that found predatory behaviour increased with temperature in *G. pulex* parasitised with *E. truttae* [93]. Although this increased food intake could suggest a higher energy demand by parasitised individuals, this was not reflected in an increase in metabolic rate, which contrasted with the findings of some other studies that found that biological effects could alter MTE scaling coefficients [68,69]. The relatively low prevalence of *E. truttae* infection indicated that the increased shredding at higher temperatures was not likely to have a significant ecosystem effect, especially as the shredding rates increased most at temperatures at which survival was reduced through species’ thermal limits being exceeded. No difference was found between amphipod species in temperature sensitivity of metabolic rates, which may reflect similarities in these relatively closely related species. Although previous research has found conspecifics with larger body sizes to be more negatively impacted by temperature increases, most likely due to limitations to metabolic processes or supply of oxygen to tissues, these *Gammaridae* family members do not appear to significantly differ in metabolic temperature sensitivity [80]. While this study used alder as a food source for shredders, the results were similar to previous experiments using sycamore, with *D. villosus* shredding rates lower than those of *G. pulex* [61]. It has been suggested that rates of shredding can be determined by leaf nutrient quality, but the activation energy results in this experiment for both species of amphipod were similar to those found in the experiment using sycamore, a lower quality resource than alder [61].

Temperature was the significant factor affecting amphipod survival, with no interaction with species. This was in accord with some previous studies [94], whereas in other studies, the native *G. pulex* was found to be slightly more tolerant of high temperatures than *D. villosus* [95]. Mortality may have been due to cumulative deleterious impacts on mitochondria as a previous experiment found that temperatures of 30 °C and above limited ATP production in *G. pulex* [96]. Population densities are also influenced by the effect of temperature on fecundity and development. Data are lacking for *D. villosus*, but research on other amphipod species indicates that higher temperatures generally decrease brood size and also decrease the development time [97,98,99]. Additional research is required to investigate how this would alter the population size and structure of both native and invasive amphipods as impacts of temperature may be amplified or lessened by changes in density over time. No effect on survival was exerted by parasitism in *G. pulex*. A low virulence would be adaptive for the acanthocephalan as parasites are transmitted from the intermediate invertebrate host when it is predated by the definitive host; hence, the survival of the intermediate host is vital to allow for the trophic transmission of the parasite.

Parasitism did not affect the metabolic rate of the host, in contrast with the findings of Labaude et al., who reported an increased metabolic rate in *G. pulex* infected with the acanthocephalan *P. laevis* [47]. The relatively large size of *E. truttae* meant it was surprising that the metabolic rate was not affected, but the results may reflect an equivalence in metabolic activity for *G. pulex* and *E. truttae* tissues by mass as well as selection on the parasites to trade off energetically and metabolically costly processes such as growth with the survival of their host [100]. The parasitism of *G. pulex* did not change the temperature scaling of the shredding behaviour, with a similar activation energy for amphipods parasitised by *E. truttae* as unparasitised individuals. Testing a range of different parasite species may produce contrasting results, as indicated in a previous study [101]. The oxygen requirements and temperature-related mortality did not differ between the native and invasive species, suggesting that the invasive species was not better adapted to climate change warming or an increasing frequency of high-temperature extremes.

The impact of the invasive *D. villosus* on resource processing depends on its leaf shredding capability relative to the native species that it replaces, its predation of native shredders to alter the shredder community structure and on its relative abundance in an invaded location. *D. villosus* typically reaches much higher densities in their invaded range compared with the densities reported for *G. pulex* [23,72] The RIP metric has previously been used to predict the potential ecological impact of predation by invasive species [71,102], including *D. villosus*, which was found to have a higher predatory impact on native shredders than the native *G. pulex* [103]. Here, we used the RIP metric to explore the potential impact of species and temperature on amphipod shredding. Although per capita shredding was higher for *G. pulex* than *D. villosus* at all temperatures, the higher abundance of this invader led to higher RIP scores for *D. villosus* at all temperatures; hence, the replacement of the native by the invasive amphipod was predicted to drive an increase in shredding. This could be interpreted as a positive effect on the ecosystem function, with a higher rate of shredding leading to a faster accumulation of amphipod biomass and a greater rate of FPOM provisioning for the ecosystem. However, the high density of invaders compared with natives may lead to the exhaustion of the resource in sites with relatively low leaf detritus levels. In addition, the flexible feeding habits of *D. villosus* could affect the community structure as competition for resources may lead to the increased predation of macroinvertebrates and fish eggs as the detrital resource reduces, the *D. villosus* population matures and its trophic level increases [104,105]. The high density of invasive shredders may also increase the nutrient load of affected water courses, with an increase in FPOM leading to possible eutrophication in freshwater systems with high seasonal allochthonous inputs [106]. Such increases in the nutrient load can alter planktonic communities, leading to algal blooms and cascading impacts to food webs [107].

## 5. Conclusions

The results demonstrated that an interaction of the stressors of temperature, invasive species and parasites could affect the ecosystem process of shredding. The invasive amphipod *D. villosus* had a lower per capita rate of shredding than the native *G. pulex*, but existed in higher densities in the environment, which indicated that its replacement of the native species could be predicted to lead to the increased processing of detritus resources. Although the rates of shredding were found to increase with temperature, it is likely that the processing of detritus will severely decline if temperatures exceed the thermal tolerances of the amphipod shredders. Once thermal optima are exceeded, it is likely that shredders will seek thermal refugia, which may limit shredding activity compared with normal foraging behaviour. Thus, shredding activity would be limited by mortality or amphipods moving away from detrital resources to seek a more favourable thermal situation [61].

The fine particulate organic matter (FPOM) produced by shredding activity supports a community of collector species, from gathering collectors (including mayfly nymphs and midge larvae) to filtering collectors such as blackfly larvae and mussels [108]. The combined effects of invasive species and increasing temperatures could have significant impacts on freshwater communities and potentially cascading effects to connected ecosystems [26].

## Figures and Tables

**Figure 1 biology-12-00830-f001:**
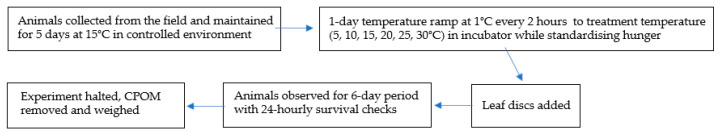
Schematic of basic experimental workflow for survival/shredding experiment.

**Figure 2 biology-12-00830-f002:**
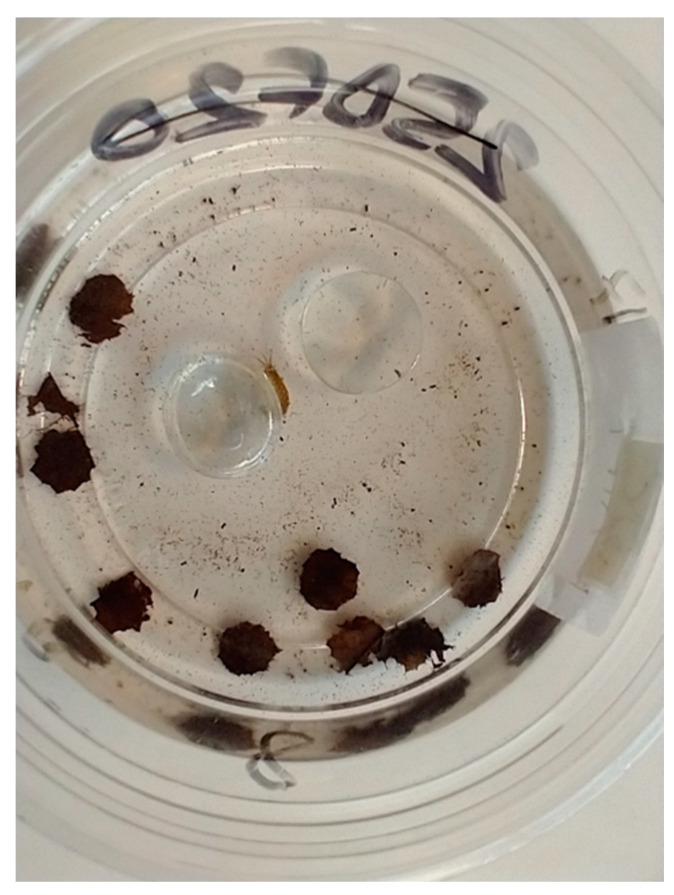
An individual *D. villosus* sheltering by a glass bead in a container with partially shredded *A. glutinosa* leaf discs. Shredded leaf matter FPOM and amphipod faeces are visible.

**Figure 3 biology-12-00830-f003:**
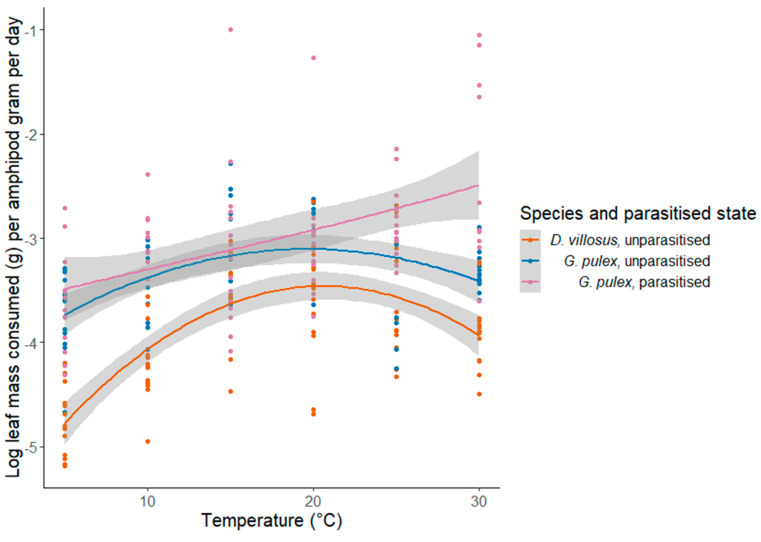
Relationship between rates of ln−transformed shredding for each amphipod treatment (species and parasitised state) and temperature, displaying standard error for Loess−smoothed curves.

**Figure 4 biology-12-00830-f004:**
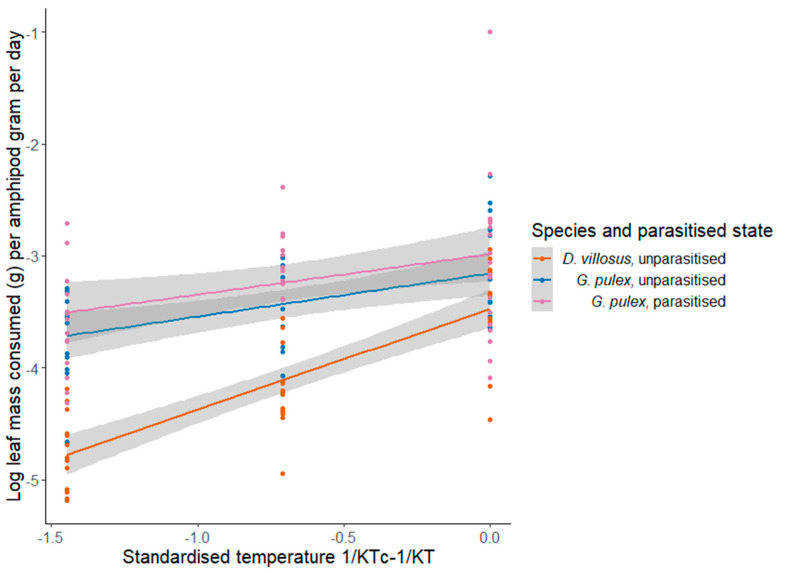
Relationship between Boltzmann−averaged standardised temperature and ln−transformed rates of shredding at 5, 10 and 15 °C for each amphipod treatment.

**Figure 5 biology-12-00830-f005:**
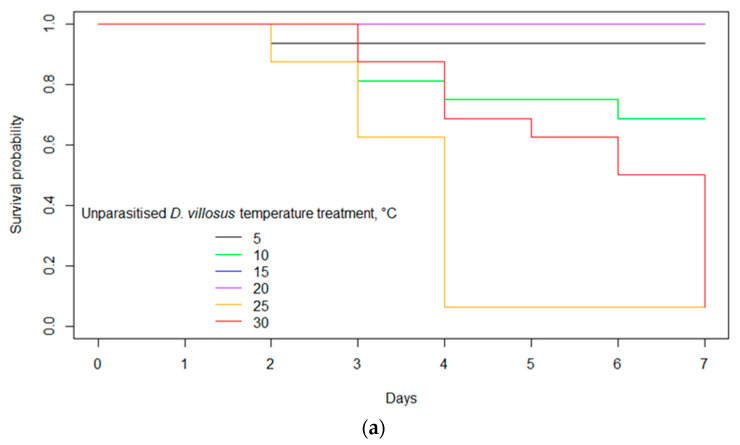
(**a**) Kaplan Meier plot of survival by temperature treatment of unparasitised *Dikerogammarus villosus. (***b**) Kaplan Meier plot of survival by temperature treatment of unparasitised *G. pulex*. *(***c**) Kaplan Meier plot of survival by temperature treatment of *G. pulex* parasitised by *E. truttae*.

**Figure 6 biology-12-00830-f006:**
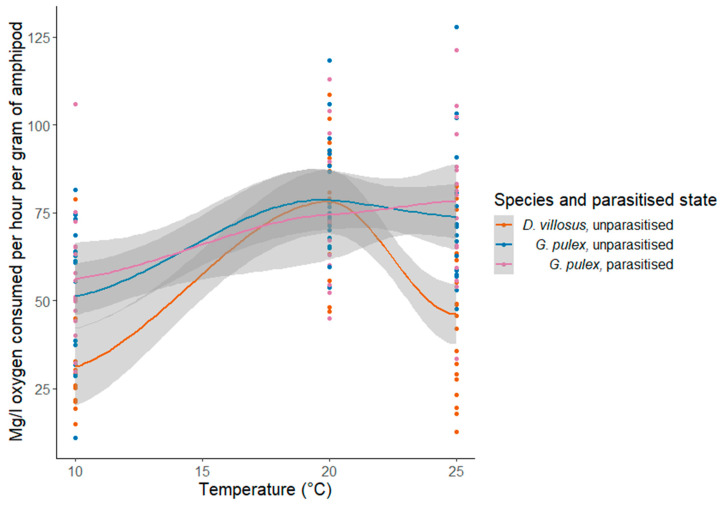
Oxygen consumption for parasitised and unparasitised *Gammarus pulex* and unparasitised *Dikerogammarus villosus* at 10, 20 and 25 °C, displaying standard error for Loess-smoothed curves.

**Figure 7 biology-12-00830-f007:**
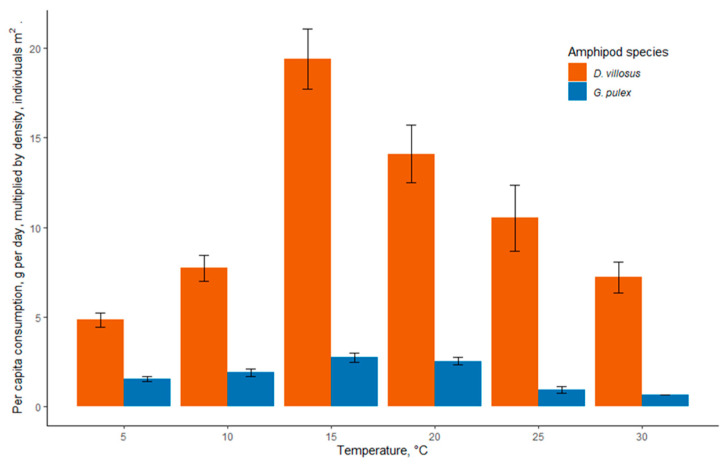
Relative impact potential of *Dikerogammarus villosus* and *Gammarus pulex* at different temperatures calculated by multiplying per capita consumption of leaf detritus by amphipod density at field locations. Error bars represent ± 1 standard error.

**Table 1 biology-12-00830-t001:** Shredding rates of each amphipod (species and parasitised state) and temperature treatment.

Amphipod Treatment	Temperature (°C)	Mean Shredding Rate (g Leaf Eaten Amphipod g^−1^ day^−1^)	St. Dev. Shredding Rate
*G. pulex* unparasitised	5	0.0263	0.0083
*G. pulex* infected with *E. truttae*	5	0.0301	0.0153
*D. villosus*	5	0.0093	0.0033
*G. pulex* unparasitised	10	0.0337	0.0127
*G. pulex* infected with *E. truttae*	10	0.0498	0.0151
*D. villosus*	10	0.016	0.0059
*G. pulex* unparasitised	15	0.0479	0.0233
*G. pulex* infected with *E. truttae*	15	0.0644	0.0793
*D. villosus*	15	0.0343	0.0111
*G. pulex* unparasitised	20	0.0548	0.0149
*G. pulex* infected with *E. truttae*	20	0.063	0.0636
*D. villosus*	20	0.0327	0.015
*G. pulex* unparasitised	25	0.0247	0.0128
*G. pulex* infected with *E. truttae*	25	0.0626	0.023
*D. villosus*	25	0.0325	0.0182
*G. pulex* unparasitised	30	0.036	0.0072
*G. pulex* infected with *E. truttae*	30	0.135	0.116
*D. villosus*	30	0.021	0.008

**Table 2 biology-12-00830-t002:** Regression parameters for ln-transformed shredding rates by Boltzmann−averaged standardised temperature for each amphipod treatment.

Amphipod Treatment	Intercept ± 1 SE	Multiplier 95% CI	*p*-Value	R^2^
*G. pulex* unparasitised	−3.16 ± 0.10	0.39 (0.17, 0.60)	<0.001	0.24
*G. pulex* parasitised	−2.99 ± 0.12	0.36 (0.09, 0.63)	<0.001	0.14
*D. villosus*	−3.47 ± 0.09	0.90 (0.72, 1.09)	<0.001	0.69

**Table 3 biology-12-00830-t003:** Metabolic rate results comparing *Dikerogammarus villosus* with unparasitised *Gammarus pulex* and *G. pulex* parasitised by *Echinorhynchus truttae* at temperatures between 10 and 25 °C.

Amphipod Treatment	Temperature (°C)	Mean Metabolic Rate (mg/L Oxygen Consumed Hour^−1^ Amphipod Gram^−1^)	St. Dev. Metabolic Rate
*G. pulex* unparasitised	10	51.2	20.5
*G. pulex* infected with *E. truttae*	10	56.2	18.1
*D. villosus*	10	31.2	16.9
*G. pulex* unparasitised	20	78.6	15.9
*G. pulex* infected with *E. truttae*	20	74.4	23.0
*D. villosus*	20	78.1	17.2
*G. pulex* unparasitised	25	73.8	21.3
*G. pulex* infected with *E. truttae*	25	78.3	22.5
*D. villosus*	25	46.1	21.7

**Table 4 biology-12-00830-t004:** Relative impact potential (RIP) results by temperature treatment comparing the invasive *Dikerogammarus villosus* with the native *Gammarus pulex*. RIP scores > 1 indicate a predicted impact of the invasive species compared with the native species in leaf shredding rate.

Temperature (°C)	Mean RIP	95% Confidence Interval		% Probability RIP > 1	% Probability RIP > 10
		Lower Limit	Upper Limit		
5	4.82	0.08	29.83	61.0	10.8
10	6.80	0.10	42.31	68.6	15.4
15	11.07	0.16	69.42	78.2	23.9
20	8.01	0.12	49.92	72.3	18.1
25	20.85	0.24	133.94	86.0	36.3
30	7.56	0.13	46.49	72.3	17.4

## Data Availability

Access to experimental data is available online at: https://osf.io/akjdc/?view_only=66f1243f36054cc182d5852f251a7b99 (accessed on 30 April 2023).

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
