# Peer review of "Biological Invasions Affect Resource Processing in Aquatic Ecosystems: The Invasive Amphipod Dikerogammarus villosus Impacts Detritus Processing through High Abundance Rather than Differential Response to Temperature"

_biology, 2023, doi:10.3390/biology12060830_

Round 1

Reviewer 1 Report

The authors present a study on the thermal response of metabolic and feeding rates in native and invasive species of amphipods. I found the study interesting and noteworthy.  However, I have some general and specific comments with the present manuscript which I believe that the author can adequately address

The core of the study is mainly based on the two main organism characteristics, namely metabolic rate and resource use, as well as their thermal dependence. The authors investigated these traits to disentangle the effects of other conditions, such as parasites and invasions. However, I believe that providing a stronger theoretical (see for example Brown et al. 2004 “Toward a metabolic theory of ecology”) and empirical background on metabolic and feeding rates (see also Shokri et al. 2022 “Metabolic rate and climate change across latitudes: evidence of mass-dependent responses in aquatic amphipods”), how they vary with size and temperature, would help readers better understand the study and the reasons for investigating these species-specific traits. Therefore, I suggest that the authors include a more comprehensive literature review on the relationship between metabolic rate, resource use, and thermal response. This review could include a discussion of existing theories and models that explain how these traits are interrelated.

I noticed that the authors did not include a separate section on data analysis in their manuscript. While it's possible that this is a common editorial style for the journal, it may be helpful for the authors to present their data analysis in a more clearly defined manner/section. One suggestion would be to include a separate subsection “e.g., Data analysis” in the M&M section specifically detailing any statistical tests used to analyze the data. Concentrating this information in a separate subsection would help readers better understand the results and replicate the study in future research.

To achieve a more coherent structure, I suggest organizing the result in the following order: first, finding on metabolic rate, followed by shredding, (This is because the findings suggest that metabolic rate encourages feeding rates, with a similar trend in thermal response); next, survival; and finally, findings on invader relative impact potential. I suggest restructuring the discussion too, with a rational flow, possibly by following the findings presented in the results section. This may help the authors to present their findings in a more cohesive manner, and help readers follow the story of the study. 

- Specific comments follow:

L 29: It would be helpful if the authors could state the temperature range used in the study.

L127-132: Some information about the average min, mean and max annual temperature of the collection site?

L131: Should be ‘acclimated’ (tend to refer to acclimatised to individuals in the wild under ‘climate’, and acclimated for those in a lab setting).

L168: I'm confused about whether the survival experiment was conducted separately from the shredding experiment or whether the authors accounted for mortality during the shredding experiment. This information is important to better understand and to evaluate the potential impact of mortality on the results.

L191: It would be helpful if the authors could explicitly state in the Methods section that metabolic rate measurements were performed on different subsets of the animals than those used in the feeding trial.

L 208: Have the authors considered the oxygen solubility at different temperatures in their final calculation of individual oxygen consumption?

L 258: It would be informative if the authors could provide some additional characteristics of the tested species, such as the range and mean size.

L 259: Based on the methods section, it appears that the authors used a generalized additive to model the thermal performance curve of feeding/metabolic rates. It would be helpful if the authors could provide a summary table or estimated coefficients.

The correspondence/harmony between changes in metabolic rate and feeding rates observed in the results as a function of temperature is intriguing. However, it is also important to discuss the potential reasons why metabolic rate and feeding rates shows a downward trend at temperatures beyond 20 °C.

To help contextualize the study, here are some studies that the authors might consider:

Resource use in amphipods: 

Shokri et al. 2021 “A new approach to assessing the space use behavior of macroinvertebrates by automated video tracking”

The thermal response of metabolic and feeding rates:

Réveillon et al. 2022“Energetic mismatch induced by warming decreases leaf litter decomposition by aquatic detritivores”

Author Response

Reviewer 1

  • include a more comprehensive literature review on the relationship between metabolic rate, resource use, and thermal response. This review could include a discussion of existing theories and models that explain how these traits are interrelated. – Response: More background has been added to the introduction and some additional analysis has also been carried out (lines 328-342). The discussion also has an MTE component
  • include a separate subsection “e.g., Data analysis” in the M&M section. – Response: data analysis now split into a subsection in materials and methods (lines 270-308)
  • organizing the result in the following order: first, finding on metabolic rate, followed by shredding, (This is because the findings suggest that metabolic rate encourages feeding rates, with a similar trend in thermal response); next, survival; and finally, findings on invader relative impact potential. I suggest restructuring the discussion too, with a rational flow, possibly by following the findings presented in the results section. – Response: The focus of the study was comparing the shredding by native and invasive amphipods. No difference was found in metabolic rate between amphipod species and so it was decided to retain our focus on the shredding data.
  • L 29: It would be helpful if the authors could state the temperature range. – Response: temperature range added (line 32)
  • L127-132: Some information about the average min, mean and max annual temperature of the collection site? Response: Information on temperature data added (lines 143-148)
  • L131: Should be ‘acclimated’ (tend to refer to acclimatised to individuals in the wild under ‘climate’, and acclimated for those in a lab setting). – Response: acclimatized changed to acclimated throughout (line 239)
  • L168: I'm confused about whether the survival experiment was conducted separately from the shredding experiment or whether the authors accounted for mortality during the shredding experiment. – Response: text added to materials and methods to clarify (lines 177-179)
  • metabolic rate measurements were performed on different subsets of the animals than those used in the feeding trial. - Response: Explanatory text added (line 225)
  • L 208: Have the authors considered the oxygen solubility at different temperatures in their final calculation of individual oxygen consumption? – Response: Oxygen solubility was not accounted for. Methods used were comparable to Kenna et a, 2017 and Yvon-Durrocher et al., 2012, allowing for comparison.
  • L 258: It would be informative if the authors could provide some additional characteristics of the tested species, such as the range and mean size. – Response: Size data added for each experiment (lines 184-185, 229-230)
  • L 259: Based on the methods section, it appears that the authors used a generalized additive to model the thermal performance curve of feeding/metabolic rates. It would be helpful if the authors could provide a summary table or estimated coefficients. Summary table to be added to supplementary material as soon as possible
  • to discuss the potential reasons why metabolic rate and feeding rates shows a downward trend at temperatures beyond 20 °C. check ref 85. Response: discussion added (lines 430-432).

Reviewer 2 Report

Review

Paper title: Biological invasions affect resource processing in aquatic ecosystems; the invasive amphipod Dikerogammarus villosus impacts detritus processing through high abundance rather than differential response to temperature

The introduction and transfer of invasive alien species between continents, regions, and nations has often had a significant impact on the aquatic and terrestrial ecosystems of the recipient countries. Policymakers and stakeholders are becoming increasingly aware of the threats posed by biological invasions to human health, economic output, ecosystem services, and biodiversity as evidence of the magnitude of the problem mounts. Dikerogammarus villosus, commonly known as the killer shrimp, is native to the Ponto-Caspian region and has invaded many European countries in recent years through a combination of natural and human-mediated dispersal. The killer shrimp is currently present in at least 17 European countries and is expected to continue to spread throughout Europe and eventually into North America. The most recent invasion in Europe is in Great Britain, where it was first detected in September 2010 in the Great Ouse River basin in eastern England. The presence of this species in these areas is of particular concern as it is an omnivorous predator known to have a dramatic impact on the native fauna of invaded ecosystems, including fish stocks, and for which there are no effective management or eradication options. For the first time, the authors conducted a laboratory study comparing metabolic and shredding rates between invasive amphipods (Dikerogammarus villosus) and native amphipods (Gammarus pulex) parasitized or unparasitized by a common acanthocephalan, Echinorhynchus truttae, at different temperatures. The authors found that higher abundances of the invader led to higher relative impact scores and predicted an increase in shredding following replacement of the native by the invasive amphipod. They concluded that this is positive for ecosystem function, but negative effects may exist at sites with relatively low levels of leaf detritus. These results may have important implications for further monitoring, management and conservation in the region.

All these reasons explain the relevance of the paper by Benjamin Pile and co-authors submitted to "Biology".

General scores.

The data presented by the authors are original and significant. The study is correctly designed and the authors used appropriate sampling methods. In general, statistical analyses are performed with good technical standards. The authors conducted careful work that may attract the attention of a wide range of specialists focused on biological invasions.

Recommendations.

In the Materials and methods, the authors should provide data on the size, sex and reproductive status of each amphipod species used in the experiments.

The sampling period is not indicated, while seasonal changes in physiology may occur, reflecting life history traits. In addition, the authors stated that changes in the trophic level of Dikerogammarus villosus occur after maturation (L 383-384). Thus, size distribution data are needed for both the experimental amphipods and their wild counterparts to assess the applicability of the experimental data to natural conditions.

In the "Results" (Sections 3.1 and 3.3), the authors should provide mean values for each measured parameter, as in the current form, only curves are presented, from which the reader cannot obtain real values for comparison.

The resolution and font size in Figures 2 and 4 need to be increased.

L 316-318. Why did the authors not compare the RIP values between the two amphipod species?

Specific remarks.

All the Latin names should be italicized.

L 109. Consider replacing “how survival and shredding differs” with “how survival and shredding differ”

L 161. Consider replacing “with 250 ml aged tap water” with “with 250 ml of aged tap water”

L 224. Consider replacing “from Warren et al (2021)” with “from Warren et al. [76]”

Author Response

Reviewer 2

  • data on the size, sex and reproductive status of each amphipod species used in the experiment. – Response: explanatory text added to materials and methods section (lines 179-185,226-230)
  • The sampling period is not indicated, while seasonal changes in physiology may occur, reflecting life history traits. In addition, the authors stated that changes in the trophic level of Dikerogammarus villosus occur after maturation. – Response: Detail added to materials and methods on amphipod selection in respect to maturity in materials and methods section (lines 179-185,226-230)
  • provide mean values for each measured parameter, as in the current form, only curves are presented, from which the reader cannot obtain real values for comparison. – Response: Tables of data added (lines 326-327, 340-341, 373-374, 382-385).
  • The resolution and font size in Figures 2 and 4 need to be increased. – Response: Font sizes increased (lines 323,369)
  • L 316-318. Why did the authors not compare the RIP values between the two amphipod species? RIP compares native as baseline to invasive. – Response: The RIP uses the native species as the baseline against which the invasive species is given a relative impact potential score (Dick et al., 2017, Invader Relative Impact Potential: a new metric to understand and predict the ecological impacts of existing, emerging and future invasive alien species)
  • All the Latin names should be italicized. – Response: italics corrected.
  • L 109. Consider replacing “how survival and shredding differs” with “how survival and shredding differ”. – Response: Text updated (line112)
  • L 161. Consider replacing “with 250 ml aged tap water” with “with 250 ml of aged tap water”. – Response: text updated (line 186)
  • L 224. Consider replacing “from Warren et al (2021)” with “from Warren et al. [76]”. – Response: Text updated (line 266)

Reviewer 3 Report

The article is informative and contributes to the study of invasive species in donor ecosystems. The manuscript describes in detail the experimental technique, the results are described in sufficient detail, the description is accompanied by drawings that are understandable to the reader. In general, I believe that this article will be useful for ecologists, zoologists and related specialists.

I have a few notes:

1) I recommend authors to add the scheme illustrating the design of the experiment to Materials and Methods. This will make it easier for the reader to understand the process of the experiment. Since the description of the experiment is quite detailed (which is good), it will be easier for the reader to first consider the circuit in the figure.

2) At the beginning of the manuscript it was said about the impact of global warming. But in fact, the impact of global climate change, of course, is not considered in the article. In the work, an assessment of the influence of temperature was carried out, and locally. It's not the same as "global warming" though. Therefore, I recommend to remove the speculative phrases about the impact of global warming.

Although the influence of temperature on parasitism status and relative impact potential is shown, but, for example, the temperature, as the authors show, does not affect the shredding rate: “there was no significant interaction between temperature and species”.

And the negative impact of temperature increase on amphipod survival does not differ between species (p. 7).

3) Make the species names in italics throughout the text (for example, in the description of the results on page 6 onwards)

4) Authors refer to previous study for population densities. But it is better to give specific values of population density. This is especially necessary to understand the relative impact potential (RIP) results.

5) In the discussion, I recommend adding information about the ecology of the invader in the natural range. Indeed, such a high abundance for a relatively young population (the authors write that the species has been known in the UK since the 2010s) may be a temporary phenomenon. It is important to show what limits the abundance in the natural range, what indicators of the population size the invasive species achieves in the natural range.

Author Response

Reviewer 3

  • scheme illustrating the design of the experiment to Materials and Methods. – Response: Schematic diagram added (line 202)
  • remove the speculative phrases about the impact of global warming. – Response: A mention of global warming has been removed from the abstract. I have checked the rest of the manuscript and ensured that only effects of temperature are mentioned. Climate change is discussed as it is likely to lead to increasing temperatures.
  • Make the species names in italics throughout the text. – Response: text corrected
  • Authors refer to previous study for population densities. But it is better to give specific values of population density. This is especially necessary to understand the relative impact potential (RIP) results.see methods. – Response: density data added (lines 299-300)
  • In the discussion, I recommend adding information about the ecology of the invader in the natural range. Indeed, such a high abundance for a relatively young population (the authors write that the species has been known in the UK since the 2010s) may be a temporary phenomenon. See 426/7 It is important to show what limits the abundance in the natural range, what indicators of the population size the invasive species achieves in the natural range. – Response: Text added to discussion concerning post-invasion abundance and dominance of villosus invasions across mainland Europe (lines396-398).

Round 2

Reviewer 1 Report

The revised version of the manuscript appears to be fine.